# Increased Association of Pulmonary Thromboembolism and Tuberculosis during COVID-19 Pandemic: Data from an Italian Infectious Disease Referral Hospital

**DOI:** 10.3390/antibiotics11030398

**Published:** 2022-03-16

**Authors:** Virginia Di Bari, Gina Gualano, Maria Musso, Raffaella Libertone, Carla Nisii, Stefania Ianniello, Silvia Mosti, Annelisa Mastrobattista, Carlotta Cerva, Nazario Bevilacqua, Fabio Iacomi, Annalisa Mondi, Simone Topino, Delia Goletti, Enrico Girardi, Fabrizio Palmieri

**Affiliations:** National Institute for Infectious Diseases “Lazzaro Spallanzani” IRCCS, 00149 Rome, Italy; virginia.dibari@inmi.it (V.D.B.); maria.musso@inmi.it (M.M.); raffaella.libertone@inmi.it (R.L.); carla.nisii@inmi.it (C.N.); stefania.ianniello@inmi.it (S.I.); silvia.mosti@inmi.it (S.M.); a.mastrobattista@inmi.it (A.M.); carlotta.cerva@inmi.it (C.C.); nazario.bevilacqua@inmi.it (N.B.); fabio.iacomi@inmi.it (F.I.); annalisa.mondi@inmi.it (A.M.); simone.topino@inmi.it (S.T.); delia.goletti@inmi.it (D.G.); enrico.girardi@inmi.it (E.G.); fabrizio.palmieri@inmi.it (F.P.)

**Keywords:** tuberculosis, pulmonary tuberculosis, pulmonary thromboembolism, COVID-19, Italy

## Abstract

Pulmonary thromboembolism (PTE) has been associated with tuberculosis (TB), but the true incidence is unknown. The aim of our study was to retrospectively evaluate the PTE prevalence in TB patients hospitalized at the National Institute for Infectious Diseases L. Spallanzani during the January 2016–December 2021 period. Retrospective data collection and evaluation were conducted. Among 1801 TB patients, 29 (1.61%) exhibited PTE. Twenty (69%) had comorbidities; eleven (37.9%) had predisposing factors for PTE. Nineteen (65.5%) had extensive TB disease. The commonest respiratory symptoms were cough (37.9%), dyspnea (31%), chest pain (10.3%), and hemoptysis (6.9%). Twenty-five (86.2%) had elevated serum D-dimer levels. An increased prevalence of PTE from 0.6% in the pre-COVID-19 pandemic period to 4.6% in the pandemic period was found. Acute respiratory failure and extensive TB disease increased significantly in the pandemic period. The increase in PTE could be explained by the increased severity of TB in patients in the pandemic period and by increased clinical suspicion and, consequently, increased requests for D-dimer testing, including in patients with non-COVID-19 pneumonia. Patients with extensive pulmonary disease are at high risk of developing PTE. Clinicians should be aware of this potentially life-threatening complication of TB, and patients should receive a thromboembolism risk assessment.

## 1. Introduction

Tuberculosis (TB) is the leading and deadliest infectious disease in the world. Globally, 7.1 million new TB diagnoses and 1.4 million deaths were estimated in 2019 [1]. In Italy, 3346 TB cases were reported in 2019, with a notification rate of 5.5% per 100,000 population [2].

Clinical TB manifestations vary depending on localization. General symptoms can include weakness, unintentional weight loss and decreased appetite, fever, and night sweats. Other symptoms depend on TB localization; with pulmonary localization, symptoms can include persistent cough and sputum production, chest pain, and hemoptysis. Other common complications of pulmonary TB can include pneumothorax or fibrothorax, atelectasis or bronchiectasis, and lung destruction [3].

Pulmonary thromboembolism (PTE) is a disease caused by thromboembolism blockage of the pulmonary artery or its branches, and it is usually associated with prolonged immobilization and known comorbidities [4,5]. PTE is also often associated with various infectious diseases [6,7,8]. A recent population-based case–control study showed that any infections, such as pneumonia and symptomatic urinary tract infection, were independent risk factors for PTE [9].

Recently, there has been growing evidence supporting the association between active TB and thrombosis [10,11]. A transitory hypercoagulability state, venous stasis, and endothelial dysfunction (Virchow’s triad) seem to play a central role. The use of rifampicin, an effective non-specific inducer of hepatic cytochrome P450, has also been implicated as a risk factor for VTE in patients with active TB [12,13,14].

Previous studies that assessed PTE prevalence in TB patients reported values ranging from 0.6% [14] to 3.5% [15]. The true incidence may, however, be higher because of non-specific or asymptomatic clinical presentations of PTE. Furthermore, PTE and TB share symptoms and signs such as cough, dyspnea, chest pain, and hemoptysis, which can hinder early recognition of PTE, especially in patients with extensive TB. To our knowledge, no published data report the PTE prevalence in TB patients in Italy. The aim of this study was to retrospectively evaluate PTE prevalence in a cohort of patients hospitalized for pulmonary TB at an Italian referral hospital, the National Institute for Infectious Diseases L. Spallanzani (INMI), during the period from 1 January 2016 to 31 December 2021.

## 2. Results

From January 2016 to December 2021, 1801 cases of TB were consecutively admitted at the INMI hospital. Of these, 29 (1.61%) patients exhibited PTE at admission or during the hospital stay: 2 (0.67%) cases out of 299 TB patients admitted at INMI in 2016, 1 (0.31%) case out of 326 in 2017, 2 (0.56%) cases out of 358 in 2018, 3 (0.83%) cases out of 362 in 2019, 9 (3.64%) cases out of 247 in 2020, and 12 (5.74%) cases out of 209 in 2021. Out of 29 cases, 3 (10.3%) were lobar PTE, 16 (55.2%) were segmental, and 10 (34.5%) were sub-segmental PTE. The median age was 50 (interquartile range: 39.5–65.5), and all 29 cases with PTE were microbiologically confirmed (Table 1). All TB strains were fully drug-sensitive, with the exception of two multidrug-resistant (MDR) strains. Patients were treated for TB according to the institutional protocol, drawn up following WHO TB guidelines, and for acute pulmonary embolism according to the European Society of Cardiology (ESC) guidelines [16,17]. Baseline and clinical characteristics are shown in Table 1. Twenty (69%) patients had single or multiple comorbidities, and eleven (37.9%) had one or more predisposing factors for PTE at the time of hospital admission. Nineteen patients (65.5%) had extensive TB disease. Three (10%) patients had a positive color Doppler examination indicating concomitant deep vein thrombosis (DVT).

Eight patients were hospitalized before the COVID-19 pandemic (2016–2019), while twenty-one patients were hospitalized during the pandemic period (2020–2021). All 21 patients hospitalized in 2020–2021 were tested for SARS-CoV-2 infection, and one was positive. No clinical features were found to be significantly different between 2020–2021 and 2016–2019 periods except for acute respiratory failure and extensive TB disease (57.1% vs. 12.5% and 71.4% vs. 25.0%, respectively). No significant differences in socio-demographic factors, comorbidities, previous TB, extrapulmonary TB, or outcome were found between 2020–2021 and 2016–2019 periods.

Elevated D-dimer levels were observed in 25/29 (86.2%) patients (median: 6756; range 707–35,326). For patients with elevated D-dimer values, the median time elapsed between D-dimer testing and positive computed tomography pulmonary angiogram (CTPA) was 1 day (range: 1–4). The average stay in the hospital lasted 36.1 days (range: 3–119). Twenty-seven (93.1%) patients were discharged, and two (6.9%) patients died.

## 3. Discussion

In this retrospective study, among 1801 TB cases admitted at INMI during the 6-year period, 29 patients (1.61%) exhibited PTE at admission or during the hospital stay.

Globally, venous thromboembolism (VTE), comprising deep vein thrombosis (DVT) and PTE, is the third most frequent acute cardiovascular syndrome behind myocardial infarction and stroke [16]. Empirical estimates of PTE incidence rates in Italy, based on European cohort studies, have ranged from 0.189 to 0.42 per 1000 in the general population [18,19]. Patients with TB are predisposed to the development of thromboembolism. Available data confirm that the burden of PTE in people with active TB is high compared to the general population, suggesting that patients with active TB are at high risk of developing PTE. In a recent paper, 0.6% of 7905 retrospectively evaluated patients were diagnosed with both TB and PTE in Korea [14]. A large retrospective study conducted in the USA in 2014 found that the prevalence of PTE in patients with active TB was 0.27% [11]. In a recent systematic review evaluating 16,190 patients, the global prevalence of PTE in patients with active TB was 5.8% [15]. Small differences in prevalence among published data could be explained by racial differences in PTE risk factors, regional differences in adopted protocols for ruling out PTE, and under-reporting of PTE diagnoses after hospital discharge [20,21]. Our data are consistent with these and consistent with other published data; in our cohort, most patients were smear-positive on sputum or BAL [11].

The extent of PTE is commonly expressed by indicating the anatomic level of the most proximal vessel affected by a clot. Conventional classification divides the proximal extension of the clot into four levels: the main pulmonary arteries, lobar arteries, segmental, and subsegmental vessels [22]. In our cohort, most PTE cases were central (lobar or segmental), while one-third of cases were peripheral PTE. PTE located centrally in the pulmonary circulation can be detected by CTPA with a high positive predictive value (PPV). The PPV decreases in peripheral PTE, and the utilization of CTPA may result in the overdiagnosis of PTE [23].

Lung scintigraphy with planar ventilation/perfusion scan can increase specificity in PTE diagnosis, and it is also proposed to be superior to CTPA in cases with other underlying lung diseases that prevent the diagnosis of PTE with CTPA [24]. Unfortunately, we could not consider the planar lung scan technique, since TB patients must be placed in respiratory isolation, and in our hospital, only computed tomography (CT) is located in an airborne infection isolation room. Since no specific diagnostic algorithms for TB patients are available, we decided to confirm clinical and laboratory suspicion of PTE by CPTA according to ESC guidelines.

The occurrence of PTE in TB patients can be characterized as the result of the inflammation state triggered by TB, which in turn determines hypercoagulability, resulting in thromboembolism [25].

The exact underlying mechanism is yet unclear, but it is described as multifactorial. The three components of Virchow’s triad—transitory hypercoagulability state, venous stasis, and endothelial dysfunction—seem to play a central role [26]. Several experimental studies have shown that proinflammatory cytokines (interleukin 1, interleukin 6, and tumor necrosis factor α) are produced during the acute phase of *M. tuberculosis* infection. These cytokines induce the production of various acute-phase inflammatory proteins and coagulation factors by hepatocytes, resulting in a hypercoagulable state [27]. In vitro, *M. tuberculosis* induces the expression of tissue factor in monocytes/macrophages, which is a primary activator of the clotting cascade [13]. In vivo, patients with active pulmonary TB displayed thrombocytosis and increased fibrinogen, factor VIII, and plasminogen activator inhibitor 1 plasma levels, associated with depressed antithrombin III and protein C levels in the first month of treatment, resulting in activated coagulation and inhibited fibrinolysis [26].

PTE without DVT, so-called de novo PTE (DNPE), is a known phenomenon. DNPE is more frequently associated with factors evoking a local hypercoagulable state compared with PE + DVT cases, such as ipsilateral rib fractures, pulmonary contusion, and infections. DNPE seems to be more often peripherally located and nonfatal compared with PTE + DVT cases [28]. On the other hand, the diagnosis of concomitant DVT has been identified as an adverse prognostic factor. Patients with concomitant DVT have an increased risk of all-cause death, PTE-related death, and recurrent VTE over 3 months of follow-up [29]. In our cohort, the vast majority of cases were DNPE, while only three (10%) cases had a positive leg ultrasound examination indicating DVT. DNPE cases were predominantly peripheral (18 vs. 8 peripheral PTE); PTE + DVT cases were predominantly subsegmental. A hypercoagulable state and systemic inflammation, rather than local factors, seem to be the main causes of PTE in TB patients and could explain the high percentage of DNPE in our cohort [11].

In addition, active lymph node TB may mechanically favor local blood stasis: venous compression by retroperitoneal lymph nodes may result in proximal thrombosis, as reported in patients with extrapulmonary TB [11]. However, we did not find a significant link between extrapulmonary localization of TB and PTE, underlining a negligible role of local factors in our population.

It is known that the severity of TB positively correlates with a higher hypercoagulable state and the occurrence of thrombosis. Studies have shown that the risk of developing PE is proportional to the severity of tubercular disease, as there is a close correlation between hematological abnormalities and the severity of clinical findings of pulmonary TB. These studies revealed that hematological abnormalities are relatively more common in severe pulmonary TB [30,31]. Most of our patients had a clinical presentation of severe TB disease, as demonstrated by the high prevalence of extensive TB disease (65.5% of patients) and respiratory failure (44.8%) at admission.

Early initiation of anti-TB medications has been reported to decrease inflammation and the consequent hypercoagulable state [14]. Unfortunately, anti-TB treatment can also contribute to the risk of VTE, since rifampicin was linked to coagulation disorders that lead to VTE or disseminated intravascular coagulation in a few case reports [32]. The mechanism is thought to be related to the metabolism of procoagulant and anticoagulant proteins; an immunologic process has also been described [33]. In our cohort, no discontinuation of rifampicin was necessary due to coagulation disorders, since the prompt start of anticoagulant therapy resulted in a timely decline in D-dimer values and in the resolution of all signs and symptoms at the follow-up evaluation.

In our study, PTE cases exhibited an upward trend over time, from 0.67% of TB cases in 2016 to 5.74% in 2021. During the pandemic period, regional hospital wards and outpatient services for TB were partially interrupted in their routine activities, and in a recent retrospective study, we found a greater severity of TB clinical presentations in the last two years due to the consequent longer delays in TB diagnosis [34]. We speculate that the increase in PTE prevalence during the 2020–2021 COVID-19 pandemic period compared to the previous period could be associated with this increased severity of clinical presentations in the two most recent years. In fact, we found that the incidence of PTE increased from 0.6% (8 PTE cases/1345 patients) in the years 2016–2019 to 4.6% (21/456) in the years 2020–2021 (*p* < 0.005). Furthermore, in the 2020–2021 period vs. the 2016–2019 period, patients with acute respiratory failure (57.1% vs. 12.5%; *p* ≤ 0.05) and extensive TB disease (71.4% vs. 25.0%; *p* ≤ 0.05) were significantly more frequently diagnosed with PTE.

SARS-CoV-2 infection has been recognized as a hyperinflammatory and prothrombotic state, with PTE being one of the most common complications of COVID-19 [35], even in patients receiving prophylactic anticoagulation [36]. Since all but 1 of the 21 patients hospitalized during the 2020–2021 period tested negative for SARS-CoV-2 infection, we can exclude a correlation between PTE and COVID-19 pneumonia in almost all cases.

In our population, the commonest presenting respiratory symptoms were cough (37.9%), dyspnea (31%), chest pain (10.3%), and hemoptysis (6.9%), common symptoms of both TB and PTE. The current evidence-based approach to the diagnosis of PTE is the sequential use of different non-invasive modalities: clinical probability assessment, D-dimer measurement, and computed CPTA [18]. The use of a validated diagnostic algorithm has been found to lower healthcare costs and also to decrease the risk of complications from unnecessary tests [35]. However, despite the overwhelming evidence to support the use of diagnostic algorithms, adherence in clinical practice is poor [36]. Furthermore, patients with PTE may complain of signs and symptoms similar to those of TB, such as cough, dyspnea, chest pain, or hemoptysis. These similarities in symptoms can make it difficult to obtain an accurate assessment of the clinical (pre-test) probability of PTE in TB patients. In the vast majority of our cases, the suspicion of PTE arose from elevated D-dimer plasma levels, suggesting that routine D-dimer testing may increase the diagnosis of PTE in TB patients, despite its low positive predictive value in the general population [16].

We found a rise in the incidence of PTE from 0.6% in the 2016–2019 period to 4.6% in the 2020–2021 period, and we speculate that the increase in PTE prevalence in the past two years period could be associated with the increased severity of TB cases.

In our opinion, a further explanation is possible. During the pandemic period, INMI was converted into a COVID-19 hospital, and PTE is a frequent complication in patients with SARS-CoV-2 infection [37,38]. It is therefore possible that our recent full-time experience in the management of COVID-19 patients may have impacted our clinical practice, leading to increased clinical suspicion and, consequently, increased requests for D-dimer testing, including in non-COVID-19 patients presenting with TB. Routine D-dimer testing may have resulted in increased confidence in the diagnosis of PTE and led to the observed increase in PTE diagnoses in TB patients during the last two years.

Our findings suggest that TB patients, especially those with extended disease, should be considered at risk for PTE and should be assessed for anticoagulant prophylaxis. Although optimal strategies for PTE risk assessment and decision making on prophylaxis in TB patients are yet to be identified, panel guidelines suggest anticoagulant prophylaxis in acutely ill medical patients. However, the decision to initiate anticoagulant prophylaxis should arise from preliminary bleeding risk assessments [37]. Active gastroduodenal ulcer, prior bleeding, and low platelet count are the strongest independent risk factors at admission for bleeding in acutely ill hospitalized patients [39,40]. Structural damage and vascular compromise may represent additional risk factors for bleeding in TB patients: hemoptysis is a well-known complication of pulmonary TB, and gastrointestinal bleeding is also often described as a complication of abdominal TB [41]. These additional risk factors should therefore be considered in benefit–risk assessments before starting anticoagulant prophylaxis in TB patients.

We recognize some limitations in our study. The retrospective nature of the study did not allow us to consider additional factors potentially influencing PTE occurrence, such as genetic thrombophilic disorders. In addition, PTE incidence could be overestimated due to referral bias, as our center is an Italian TB referral hospital. Finally, the fact that the patient sample came from a single center may limit the extent to which our findings can be generalized.

## 4. Materials and Methods

### 4.1. Study Design and Participants

Retrospective data collection and evaluation were conducted to include all patients consecutively hospitalized for TB at INMI from 1 January 2016 to 31 December 2021. Data were extracted from the local TB database approved by the L. Spallanzani Institute Ethics Committee (Decision No. 12/2015). All patients provided written informed consent to the utilization of anonymized clinical data. The inclusion criteria were as follows: diagnosis of tuberculosis according to WHO guidelines [16], age above 18 years, and presence of PTE diagnosed by CPTA. Collected data included patients’ socio-demographic characteristics (i.e., sex, age, and nationality), comorbidities, and risk factors for TB (cardio- and cerebrovascular diseases, HIV infection, chronic alcoholism, metabolic disorders, malignancy, liver disease, mental disorders, anemia, kidney failure, and other respiratory diseases), and clinical and microbiological characteristics (symptoms, body mass index (BMI), drug resistance pattern, and concurrent extrapulmonary TB). Respiratory rate, heart rate, and arterial blood gas analysis values were included. Respiratory failure was defined as arterial oxygen pressure (PaO2) less than 60 mmHg (8.0 kPa) when breathing room air [42]. PTE predisposing factors, according to ESC guidelines, were also evaluated and are reported in Table 1 [16]. Data on radiological severity of TB at admission were collected: extensive (or advanced) TB disease was defined as the presence of bilateral cavitary disease or extensive parenchymal damage on chest radiography [43]. PTE was defined as the presence of a thrombus in the pulmonary vessels diagnosed by CT pulmonary angiography. According to ESC guidelines, CTPA was performed to confirm PTE diagnosis in patients with clinical suspicion or as a second-line test in patients with elevated D-dimer plasma levels (defined by a standard cut-off > 500 µg/L for patients aged ≤ 50 years or age-adjusted cut-offs for patients aged > 50 years) [16]. Padua prediction scoring was performed to assess the risk of DVT in all patients with PTE, and leg ultrasound was performed in patients with Padua score ≥ 4 or in patients with clinical evidence of DVT.

### 4.2. Statistical Analysis

Data were summarized using counts and percentages for categorical variables; medians and interquartile ranges were utilized for continuous variables. Risk factors were assessed using Fisher’s exact test for categorical variables and Mann–Whitney two-sample statistics to test differences for quantitative variables. All tests were two-sided, and *p*-values < 0.05 were considered significant. The data were also stratified by calendar period (2016–2019 vs. 2020–2021), and differences between the two periods were assessed using Mann–Whitney test for continuous measures and Fisher’s exact test for categorical data. Data were analyzed using Stata software, release 15.0 (Stata Corp, College Station, TX, USA).

## 5. Conclusions

In our experience, patients with extensive pulmonary disease are at high risk of developing PTE. Clinicians should be aware of this potentially life-threatening complication of TB, and patients should receive a thromboembolism risk assessment at admission and during their hospital stay.

The true role of D-dimer in the inflammation/infection-associated VTE diagnostic algorithm must be elucidated in further research. If validated scores for ruling out PTE in TB patients are not available, the clinician can rely on clinical suspicion and D-dimer age-adjusted values.

Given the lack of validated diagnostic algorithms or guidelines for PTE in TB patients, we suggest performing systematic measurements of D-dimers in all TB patients with extensive disease. PTE should be suspected and investigated with imaging tests whenever serum D-dimer levels are elevated.

Further research is needed to evaluate all risk factors that may increase the risk of PTE in TB patients and to develop validated diagnostic algorithms to arrive at a timely diagnosis and treatment of PTE in these patients.

## Figures and Tables

**Table 1 antibiotics-11-00398-t001:** Baseline and clinical characteristics of TB patients with PTE, January 2016−December 2021 (*n* = 29).

		2016–2019 (tot 8)	2020–2021 (tot 21)	*p*-Value
		N (%)	N (%)	
Gender	Male	6 (75)	13 (61.9)	0.507
	Female	2 (25)	8 (38.1)	
Nationality	Italian	4 (50)	7 (33.3)	0.811
	African	2 (25)	7 (33.3)	
	East European	2 (25)	6 (28.6)	
	Asian	0 (0)	1 (4.8)	
BMI (kg/m^2^)	Low (16–18.49)	3 (37.5)	11 (52.4)	0.574
	Normal (18.5–24.99)	5 (62.5)	9 (42.8)	
	High (25–29.99)	0 (0)	1 (4.8)	
Smoking	Yes	2 (25)	7 (33.3)	0.665
Comorbidities	Cardio- and cerebrovascular diseases	3 (37.5)	4 (19.0)	0.299
	Chronic alcoholism	1 (12.5)	4 (19.0)	0.677
	Metabolic disorders	2 (25)	5 (23.8)	0.947
	Malignancy	1 (12.5)	3 (14.3)	0.901
	Liver disease	1 (12.5)	2 (9.5)	0.814
	Mental disorders	1 (12.5)	1 (4.8)	0.462
	Other respiratory diseases	1 (12.5)	3 (14.3)	0.901
	HIV infection	1 (12.5)	0 (0)	0.099
	Anemia	0 (0)	5 (23.8)	0.129
	Kidney failure	0 (0)	1 (4.8)	0.53
Concurrent extrapulmonary TB	Yes	3 (37.5)	8 (38.1)	0.976
Symptoms	Fever	1 (12.5)	7 (33.3)	0.262
	Cough	2 (25)	9 (42.8)	0.376
	Dyspnea	2 (25)	7 (33.3)	0.665
	Weight loss	0 (0)	8 (38.1)	<0.05
	Hemoptysis	1 (12.5)	1 (4.8)	0.462
	Lipothymia	0 (0)	2 (9.5)	0.366
	Asthenia	0 (0)	6 (28.6)	0.09
	Vomiting	0 (0)	2 (9.5)	0.366
	Night sweats	0 (0)	2 (9.5)	0.366
	Chest pain	1 (12.5)	2 (9.5)	0.814
Microbiological findings	Sputum smear positive	5 (62.5)	16 (76.2)	0.516
	Sputum smear negative/molecular positive test	1 (12.5)	2 (9.5)	
	BAL smear positive	1 (12.5)	0 (0)	
	BAL smear negative/molecular positive test	1 (12.5)	2 (9.5)	
	Sputum smear-BAL molecular negative/culture positive	0 (0)	1 (4.8)	
Acute respiratory failure	Yes	1 (12.5)	12 (57.1)	<0.05
Extensive TB disease	Yes	2 (25)	17 (80.9)	<0.05
PTE predisposing factors	Previous VTE/PTE	2 (25)	1 (4.8)	0.11
	Autoimmune diseases	1 (12.5)	2 (9.5)	0.901
	Chemotherapy	0 (0)	2 (9.5)	0.366
	Congestive heart failure	2 (25)	2 (9.5)	0.28
	Other infections	2 (25)	5 (14.3)	0.947
	Malignancy	1 (12.5)	3(14.3)	0.901
	Paralytic stroke	1 (12.5)	1 (4.8)	0.462
	Bed rest > 3 days	1 (12.5)	0 (0)	0.099
	Diabetes mellitus	1 (12.5)	1 (4.8)	0.462
	Arterial hypertension	2 (25)	2 (9.5)	0.28
D-dimer values	Elevated	6 (75)	20 (95.2)	0.11
	Normal	2 (25)	1 (4.8)	

BMI: body mass index; HIV: human immunodeficiency virus; BAL: bronchoalveolar lavage; VTE: venous thromboembolism.

## Data Availability

All relevant data are within the manuscript. Raw data are accessible, if requested, from National Institute for Infectious Diseases “L. Spallanzani” Library to E-mail address: to biblioteca@inmi.it.

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
