# Peer review of "Increased Association of Pulmonary Thromboembolism and Tuberculosis during COVID-19 Pandemic: Data from an Italian Infectious Disease Referral Hospital"

_antibiotics, 2022, doi:10.3390/antibiotics11030398_

Round 1

Reviewer 1 Report

The Authors conducted a retrospective study in field of relation between pulmonary embolism and tuberculosis. In COVID pandemia the prevalence of PE seemed to be elevated. However, there are two week points of study, which must be elucidited.

  1. The pulmonary embolism/local thrombosis may be overdiagnosed using CT angiography. If subsegmental level is affected, perfusion scintigraphy is excluding the true PE in relatively high rate of cases. The Authors must publish, how many cases had a positive duplex US or color Doppler examination detecting the true source of VTE. The 3 cases with negatíve D-dimers (with extreme excellent negative predictive value!!!!) are suggesting the same observations. Thus, the detected PE severity (lobar-segmental-subsegmental) must be published as well.  In to many subsegmental cases the improvement in CT diagnosis may be the true cause of PE results. In COVID pandemia, the specialists were extremely aware of significance of microthrombosis leading probably to increased prevalence of PE diagnosis. Furthermore, the tuberculosis may result in local thrombosis without thrombotic source. I suggest to speak about their relation in detail using the previously recommended results.
  2. The true role of D-dimer in inflammation-infection associated VTE diagnostic algorythm must be in further research elucidated. The cost effectivity, the negative and positive predictive value may not be determined using these calculations.  Thus, the Authors may provide just suggestions using these data! Otherwise, when positive CT angiography is the gold standard, the positive predictive value and negative predictive value of D-dimer may be calculated in this population, too.

Author Response

REVIEWER 1

Comments and Suggestions for Authors

The Authors conducted a retrospective study in field of relation between pulmonary embolism and tuberculosis. In COVID pandemia the prevalence of PE seemed to be elevated. However, there are two week points of study, which must be elucidited.

Response

We thank you very much for the encouraging feedback on our manuscript.  Below the answer for your interesting and appropriate questions.

Question 1. The pulmonary embolism/local thrombosis may be overdiagnosed using CT angiography. If subsegmental level is affected, perfusion scintigraphy is excluding the true PE in relatively high rate of cases. The Authors must publish, how many cases had a positive duplex US or color Doppler examination detecting the true source of VTE. The 3 cases with negatíve D-dimers (with extreme excellent negative predictive value!!!!) are suggesting the same observations. Thus, the detected PE severity (lobar-segmental-subsegmental) must be published as well.  In to many subsegmental cases the improvement in CT diagnosis may be the true cause of PE results. In COVID pandemia, the specialists were extremely aware of significance of microthrombosis leading probably to increased prevalence of PE diagnosis. Furthermore, the tuberculosis may result in local thrombosis without thrombotic source. I suggest to speak about their relation in detail using the previously recommended results.

Response

Thank you for this suggestion.

We are aware that despite numerous guidelines and algorithms that aim to prevent overtesting with CTPA in patients at low risk of PE, there is ample evidence that CTPA is frequently inappropriately ordered [Osman M, Subedi SK, Ahmed A, et al. Computed tomography pulmonary angiography is overused to diagnose pulmonary embolism in the emergency department of academic community hospital. J Community Hosp Intern Med Perspect 2018; 8: 6–10]. 

We agree with utility of perfusion scintigraphy in confirm diagnosis of PE. Unfortunately, here we have patients with infection control issues, so we did not considered the technique, since in our hospital only CT is located in an airborne infection isolation room. Considering that no specific diagnostic algorithm for Tuberculosis (TB) patients are available, we decided to confirm clinical suspicion of PE by CPTE according to ESC guidelines, who suggest to evaluate age adjusted D-dimer value. There is an emerging consensus that TB is a risk factor for PE and moreover the value of empirical clinical judgement has been confirmed in several large series [Sanders S, Doust J, Glasziou P. A systematic review of studies comparing diagnostic clinical prediction rules with clinical judgment. PLoS One 2015]. We agree that the hypercoagulable state seen in TB infection is due to the imbalance of procoagulants and anticoagulants. These are represented by elevated levels of thrombin-antithrombin complexes, D-dimer, and fibrinogen along with reduced levels of antithrombin, protein C and S. When immune cells come in contact with TB there are increased amounts of TNF-α, IL-1, and IL-6.This results in systemic inflammation. Even if local factors may also induce the thrombosis of pulmonary arteries, no predominant link between pulmonary TB and pulmonary embolism has been established favoring the concept of a systemic inflammatory state leading to hypercoagulability over local thrombosis [Dentan, C, Epaulard, O, Seynaeve, D, Genty, C, Bosson, JL. Active tuberculosis and venous thromboembolism: association according to international classification of diseases, ninth revision hospital discharge diagnosis codes. Clin Infect Dis. 2014;58:495-501].

In our cohort, 3/29 (10.3%) showed lobar PTE, 16 (55.2%) segmental and 10 (34.5%) subsegmental PTE.  Only three (10%) cases had a positive color Doppler examination detecting the true source of VTE. In the other cases, TB infection seems to be the only cause of local thrombosis, without thrombotic source.

We accordingly modified the manuscript at:

  • page 2 lines 71-72 as follow: Three out of 29 (10.3%) were lobar PTE, 16 (55.2%) were segmental and 10 (34.5%) were subsegmental PTE.

  • page 2 lines 80-81 as follow: Three (10%) patients had a positive color Doppler examination detecting concomitant DVT.;

  • page 4 lines 119-134 as follow: The extent of PTE is commonly expressed by indicating the anatomic level of the most proximal vessel affected by a clot. Conventional classification divides the proximal extension of the clot into four levels: the main pulmonary arteries, lobar arteries, segmental, and sub-segmental vessels [Uflacker R. Atlas of Vascular Anatomy: an Angiographic Approach, 1st edn. Baltimore, MD: Lippencott Williams & Wilkins, 1997]. In our cohort, most of PTE were central (lobar or segmental) while one third of cases were peripheral PTE. PTE located centrally in the pulmonary circulation can be detected by CTPA with a high PPV. The PPV decreases in peripheral PTE and the utilization of CTPA may result in overdiagnosis of PTE [Newnham M, Turner AM. Diagnosis and treatment of subsegmental pulmonary embolism. World J Respirol 2019; 9(3): 30-34 --- Stein PD, Fowler SE, Goodman LR, Gottschalk A, Hales CA, Hull RD, et al. Multidetector computed tomography for acute pulmonary embolism. N Engl J Med. 2006;354:2317–2327]. Lung scintigraphy with planar ventilation/perfusion scan can increase specificity in PTE diagnosis and it is also proposed to be superior to CTPA in cases with other underlying lung diseases preventing the diagnosis of PTE with CTPA [Bajc M, Schümichen C, Grüning T, Lindqvist A, Le Roux PY, Alatri A, Bauer RW, Dilic M, Neilly B, Verberne HJ, Delgado Bolton RC, Jonson B. EANM guideline for ventilation/perfusion single-photon emission computed tomography (SPECT) for diagnosis of pulmonary embolism and beyond. Eur J Nucl Med Mol Imaging. 2019 Nov;46(12):2429-2451]. Unfortunately, we could not consider planar lung scan technique, since TB patients must be placed in respiratory isolation and in our hospital only CT is located in an airborne infection isolation room. Since no specific diagnostic algorithms for TB patients are available, we decided to confirm clinical and laboratory suspicion of PTE by CPTA according to ESC guidelines.

  • page 5 lines 151 – 163 as follow: PTE without DVT, so called de novo PTE (DNPE), is a known phenomenon. DNPE is more frequently associated with factors evoking a local hypercoagulable state compared with the PE + DVT cases, like ipsilateral rib fractures, pulmonary contusion, and infections. DNPE seems to be more often peripherally located and non fatal compared with PTE + DVT cases. Van Gent JM, Zander AL, Olson EJ, Shackford SR, Dunne CE, Sise CB, Badiee J, Schechter MS, Sise MJ. Pulmonary embolism without deep venous thrombosis: De novo or missed deep venous thrombosis? J Trauma Acute Care Surg. 2014 May;76(5):1270-4. On the other hand, diagnosis of concomitant DVT has been identified as an adverse prognostic factor. Patients with concomitant DVT have an increased risk of all-cause death, PTE-related death and recurrent VTE over 3 months of follow-up (Jime´nez D, Aujesky D, Dıaz G, Monreal M, Otero R, Martı D, Marin E, Aracil E, Sueiro A, Yusen RD; RIETE Investigators. Prognostic significance of deep vein thrombosis in patients presenting with acute symptomatic pulmonary embolism. Am J Respir Crit Care Med 2010;181:983991). In our cohort, the vast majority of cases were DNPE, while only three (10%) cases had a positive leg ultrasound examination detecting DVT. DNPE were predominantly (18 vs 8 peripheral PTE); PTE + DVT cases were predominantly sub-segmental. The hypercoagulable state and systemic inflammation, rather than local factors, seem to be the main causes of PTE in TB patients and could explain the high the high percentage of DNPE in our cohort. [Dentan, C, Epaulard, O, Seynaeve, D, Genty, C, Bosson, JL. Active tuberculosis and venous thromboembolism: association according to international classification of diseases, ninth revision hospital discharge diagnosis codes. Clin Infect Dis. 2014;58:495-501]

  • 7 lines 270 – 272 as follow: Padua prediction score was performed to assess the risk of DVT in all patients with PTE and leg ultrasound was performed in patients with Padua score ≥  4 or in patients with clinical evidence of VTE.

Question 2. The true role of D-dimer in inflammation-infection associated VTE diagnostic algorithm must be in further research elucidated. The cost effectivity, the negative and positive predictive value may not be determined using these calculations.  Thus, the Authors may provide just suggestions using these data! Otherwise, when positive CT angiography is the gold standard, the positive predictive value and negative predictive value of D-dimer may be calculated in this population, too.

Response

D-dimer testing is an integral part of validated algorithms for the diagnosis of deep-vein thrombosis (DVT) and pulmonary embolism (PTE). Clinical pre-test probability is used to guide further testing. If PTE is unlikely, D-dimer testing is the next step. D-dimer levels are elevated in most patients with acute thrombosis, but the levels also are increased with advanced age, after surgery, during pregnancy and the puerperium, with cancer and chronic inflammatory conditions. Therefore, D-dimer is a sensitive marker for detection of thrombosis, but it lacks specificity. Exploiting these test characteristics, a normal high-sensitivity D-dimer level in these settings helps to exclude the diagnosis of VTE/PTE. Because of the low specificity of high-sensitivity D-dimer tests, however, some patients with suspected VTE/PE may have negative results [Keller K, Beule J, Balzer JO, et al. D-Dimer and thrombus burden in acute pulmonary embolism. Am J Emerg Med. 2018;36:1613–8.]. D-dimer testing is of limited value in patients with a high pre-test probability because even if the D-dimer level is normal, the negative predictive value of the test is reduced by the high prevalence of VTE in such patients.  Because D-dimer levels increase with age, one maneuver uses an age-adjusted cut-off. The other maneuver adjusts the cut-off on the basis of the pre-test clinical probability because the prevalence of VTE is lower in patients with a low pre-test probability than in those with a moderate pre-test probability. This approach increases the specificity of D-dimer testing without compromising its sensitivity, and has been prospectively validated in patients with suspected PTE. 

PTE has similar symptoms, such as chest pain and shortness of breath, with TB. In addition to elevated D-dimer levels, whether to perform CTPA to confirm the PTE diagnosis is a dilemma for clinicians due to its high cost, time consumption, and potential risk of kidney injury. In the lack of validate scores for ruling out PTE in TB patients is available, we had to rely only on clinical suspicion and D-dimer adjusted values. Moreover, Li X, et al. found that D-dimer is an objective biomarker for predicting PE in patients with TPE [Diagnostic performance of D-dimer in predicting pulmonary embolism in tuberculous pleural effusion patients. BMC Pulm Med. 2021;21(1):177. Published 2021 May 22].

We deeply regret this important omission, but in the current state of affairs we can not calculate PPV and NPV in our cohort of patients because we have not collected D-dimer test results in the 1772 TB patients without TPE.

We accordingly modified the manuscript at page 7 lines 287-294 as follow:

The true role of D-dimer in inflammation-infection associated VTE diagnostic algorythm must be in further research elucidated. In the lack of validate scores for ruling out PTE in TB patients is available, the clinician can rely on clinical suspicion and D-dimer age-adjusted values. Given the lack of validated diagnostic algorithms or guidelines for PTE in TB patients, we suggest to perform systematic measurements of d-dimers in all TB patients with extensive disease. PTE should be suspected and investigated with imaging tests whenever serum D-dimer levels are elevated.

Reviewer 2 Report

Title: 

  • The title is misleading. Even though some of the patients were recruited during the COVID-19 era, however, only one of the patients contracted with COVID-19 infection. The association between PTB and COVID-19 and PTE were not being studied. Suggest to remove the word COVID-19 pandemic from the title.
  • Also, the increase of PTE in PTB patients I suppose is likely due to the change of the clinical practice over the years that D-dimer has become a routine – hence the rise in the cases. It has nothing to do with COVID-19 or pre-COVID era.

Methodology:

  • Suggest regression analysis to further analyse each risk factor mentioned that predispose to PTE in PTB patients.

Discussion:

  • Page 5, line 165. Yes, I agreed that SARS-CoV-2 infection aggravates the severity of PTB and vice versa. However, this is not the case in this study as almost all patients were negative for COVID-19 infection. The question now is, why is there increase in PTB severity within the recent two years compared to previous? Besides increase the number of D-dimer testing in local setting, is there any other possible explanation? Discuss along that.
  • Risk factors that predispose to increase in incidence of PTE in PTB patients. Do discuss that too along with pathophysiology of disease.

Table 1:  Suggest to display clinical characteristics in 2 columns – i.e. one column for TB with co-infection with COVID-19 vs TB in pre-COVID era. Get a p value if possible.

General:

Avoid the use of abbreviations without them being first elaborated in full form. E.g. COPD, BMI, PaO2, CT etc.

Author Response

REVIEWER 2

Comments and Suggestions for Authors

Response

We thank you very much for the encouraging feedback on our manuscript.  Below the answer for your interesting and appropriate questions.

Question 1

Title:  The title is misleading. Even though some of the patients were recruited during the COVID-19 era, however, only one of the patients contracted with COVID-19 infection. The association between PTB and COVID-19 and PTE were not being studied. Suggest to remove the word COVID-19 pandemic from the title.

Response

Thank you for this important remark. Even if the title could be misleading, we would humbly point out that the title want to refer to pandemic period and not to Sars Cov2 disease.

As stated in the manuscript in our study, PTE cases exhibit an upward trend over the time, from 0.63% of TB cases in 2016 to 7.14% in 2021. In a recent retrospective study we found a greater severity of TB clinical presentations in the last two years, due to COVID pandemic and consequent higher TB diagnostic delay. The coronavirus disease (COVID-19) pandemic has affected clinical management of tuberculosis (TB) and TB-related services.  We contributed to a multicentric survey published in 2020,  in which we demonstrated that coronavirus disease has disrupted TB services globally. Data from 33 centers in 16 countries on 5 continents showed that attendance at tuberculosis centers was lower during the first 4 months of the pandemic in 2020 than for the same period in 2019. [Migliori GB, Thong PM, Akkerman O, et al. Worldwide Effects of Coronavirus Disease Pandemic on Tuberculosis Services, January-April 2020. Emerg Infect Dis. 2020;26(11):2709-2712]

We speculate that the increasing in PTE prevalence during the COVID pandemic period 2020-2021 compared to the previous period could be associated with this increased severity of clinical presentations in the latest two years, not to the TB and SARS-CoV-2 coinfection.

[Pai, M., Kasaeva, T., & Swaminathan, S. (2022). Covid-19's Devastating Effect on Tuberculosis Care - A Path to Recovery. The New England journal of medicine, 10.1056/NEJMp2118145]

Question 2.

Also, the increase of PTE in PTB patients I suppose is likely due to the change of the clinical practice over the years that D-dimer has become a routine – hence the rise in the cases. It has nothing to do with COVID-19 or pre-COVID era.

Response

We thank you for excellent observation. You have raised an important point here. PE has similar symptoms, such as chest pain and shortness of breath, with TB. Due to the lack of validated specific algorithms for ruling out PTE probability in TB, we had to rely on clinical suspicion and D-dimer adjusted values. It’s true that clinical awareness and highest clinical suspicion has been derived from COVID experience, leading to change in clinical practice and to an increased number of CPTA and diagnosis of PE.

Question 3.

Methodology:

Suggest regression analysis to further analyses each risk factor mentioned that predispose to PTE in PTB patients.

Response

We deeply regret this important omission, but in the current state of affairs we could not perform regression analysis to further analyze each risk factor predisposing to PTE in TB patients, because we have not collected risk factor in the 1772 TB patients without TPE. However,  we have added the p-value to all predisposing factors in table one. We have also assessed the differences between the two periods and reported the p-value the in the same table 1.

Question 4.

Discussion:

Page 5, line 165. Yes, I agreed that SARS-CoV-2 infection aggravates the severity of PTB and vice versa. However, this is not the case in this study as almost all patients were negative for COVID-19 infection. The question now is, why is there increase in PTB severity within the recent two years compared to previous? Besides increase the number of D-dimer testing in local setting, is there any other possible explanation? Discuss along that.

Response

Thank you for the remark.

In the manuscript we stated that “Since all 21 patients hospitalized 167 during 2020-2021 period except one were tested negative for SARS-CoV-2 infection, we can exclude in almost all cases a correlation between PTE and COVID-19 pneumonia”

Coronavirus disease has disrupted tuberculosis services globally. Data from 33 centres in 16 countries on 5 continents showed that attendance at tuberculosis centres was lower during the first 4 months of the pandemic in 2020 than for the same period in 2019

We speculate that the increasing in PTE prevalence during the COVID pandemic period 2020-2021 compared to the previous period could be associated with this increased severity of clinical presentations in the latest two years.

It is known that severity of TB positively correlates with a higher hypercoagulable state and the occurrence of thrombosis. Studies have shown that the risk of developing PE is proportional to the severity of tubercular disease as there is a close correlation between the hematological abnormalities and the severity of clinical findings of pulmonary TB. The studies have revealed that hematological abnormalities are relatively more common in severe pulmonary TB. Most of our patients had clinical presentation of severe TB disease, as demonstrated by the high prevalence of extensive TB disease (65.5% of patients) and respiratory failure (44.8%) at the admission. Besides increase the number of D-dimer testing, we raised clinical suspicion on PE on based on the experience that has been derived from COVID experience.

We accordingly modified manuscript at page 5 line 187 – 190 as follow: During the pandemic period regional hospital wards and outpatient services for TB were partially interrupted in their routine activities and in a recent retrospective study we found a greater severity of TB clinical presentations in the last two years, due to and the consequent higher TB diagnostic delay [29].

Question 5. Risk factors that predispose to increase in incidence of PTE in PTB patients. Do discuss that too along with pathophysiology of disease.

Response

Thank you for your pointing out. Our cases show that PTE may complicate severe pulmonary tuberculosis and that these events occur at presentation or later in the course of the disease. Actually, tuberculosis is a disease with a wide variety of clinical presentations and recently, the association between inflammation, haemostatic changes and a hypercoagulable state has been established. Robson et al. research study suggested that elevated plasma fibrinogen, impaired fibrinolysis coupled with decreased levels of antithrombin III and reactive thrombocytosis appeared to favour the development of PTE/DVT in pulmonary TB [Robson SC, White NW, Aronson I, Woolgar R, Goodman H, Jacobs P. Acute-phase response and the hypercoagulable state in pulmonary tuberculosis. Br J Haematol. 1996;93:943–949. doi: 10.1046/j.1365-2141.1996].

Table 1 show the distribution of PTE risk factors in the two periods before and after pandemic COVID 19. No one of the predisposing risk factors was found significantly different between the two periods.

Like other infectious diseases, TB can cause thrombosis by various mechanisms such as local invasion, venous compression. Because VTE can be fatal, it is crucial to suspect it to perform an early diagnosis and initiate prompt treatment. For this reason, patients that respond poorly to TB treatment, who have other predisposing factors and those in need of a prolonged stay in hospital, should be carefully monitored [Ambrosetti M, Ferrarese M, Codecasa L, Besozzi G, Sarassi A, Viggiani P, Migliori G. Incidence of Venous Thromboembolism in Tuberculosis Patients. Respiration. 2006;73:396]

Question 6.

Table 1:  Suggest to display clinical characteristics in 2 columns – i.e. one column for TB with co-infection with COVID-19 vs TB in pre-COVID era. Get a p value if possible.

Response

Thank you for important question. As there was only one patient with TB/SARS-CoV-2 coinfection, we calculated differences in variables between the two periods and modified manuscript at page 3, table 1, as follow:

Table 1. Baseline and clinical characteristics of TB patients with PTE, January 2016−December 2021 (n = 29).

2016-2019 (tot 8)

2020-2021 (tot 21)

p-Value

N (%)

N (%)

Gender

Male

Female

6 (75)

2 (25)

13 (61.9)

8 (38.1)

0.507

Nationality

Italian

African

East European

Asian

4 (50)

2 (25)

2 (25)

0 (0)

7 (33.3)

7 (33.3)

6 (28.6)

1 (4.8)

0.811

BMI (kg/m2)

Low (16-18,49)

Normal (18.5-24,99)

High (25-29,99)

3 (37.5)

5 (62.5)

0 (0)

11 (52.4)

9 (42.8)

1 (4.8)

0.574

Smoking

Yes

2 (25)

7 (33.3)

0.665

Comorbidities

Cardio- and cerebrovascular diseases

Chronic alcoholism

Metabolic disorders

Malignancy

Liver disease

Mental disorders

Other respiratory diseases

HIV infection

Anemia

Kidney failure

3 (37.5)

1 (12.5)

2 (25)

1 (12.5)

1 (12.5)

1 (12.5)

1 (12.5)

1 (12.5)

0 (0)

0 (0)

4 (19.0)

4 (19.0)

5 (23.8)

3 (14.3)

2 (9.5)

1 (4.8)

3 (14.3)

0 (0)

5 (23.8)

1 (4.8)

0.299

0.677

0.947

0.901

0.814

0.462

0.901

0.099

0.129

0.530

Concurrent extrapulmonary TB

Yes

3 (37.5)

8 (38.1)

0.976

Symptoms

Fever

Cough

Dyspnea

Weight loss

Hemoptysis

Lipothymia

Asthenia

Vomiting

Night sweats

Chest pain

1 (12.5)

2 (25)

2 (25)

0 (0)

1 (12.5)

0 (0)

0 (0)

0 (0)

0 (0)

1 (12.5)

7 (33.3)

9 (42.8)

7 (33.3)

8 (38.1)

1 (4.8)

2 (9.5)

6 (28.6)

2 (9.5)

2 (9.5)

2 (9.5)

0.262

0.376

0.665

<0.05

0.462

0.366

0.090

0.366

0.366

0.814

Microbiological findings

Sputum smear positive

Sputum smear negative/molecular positive test

BAL smear positive

BAL smear negative/molecular positive test

Sputum smear-BAL molecular negative/culture positive

5 (62.5)

1 (12.5)

1 (12.5)

1 (12.5)

0 (0)

16 (76.2)

2 (9.5)

0 (0)

2 (9.5)

1 (4.8)

0.516

Acute respiratory failure

Yes

1 (12.5)

12 (57.1)

<0.05

Extensive TB disease

Yes

2 (25)

17 (80.9)

<0.05

PTE predisposing factors

Previous VTE/PTE

Autoimmune diseases

Chemotherapy

Congestive heart failure

Other infections

Malignancy

Paralytic stroke

Bed rest >3 days

Diabetes mellitus

Arterial hypertension

2 (25)

1 (12.5)

0 (0)

2 (25)

2 (25)

1 (12.5)

1 (12.5)

1 (12.5)

1 (12.5)

2 (25)

1 (4.8)

2 (9.5)

2 (9.5)

2 (9.5)

5 (14.3)

3(14.3)

1 (4.8)

0 (0)

1 (4.8)

2 (9.5)

0.110

0.901

0.366

0.280

0.947

0.901

0.462

0.099

0.462

0.280

D-dimer values

Elevated

Normal

6 (75)

2 (25)

20 (95.2)

1 (4.8)

0.110

BMI: body mass index; BAL: bronchoalveolar-lavage; VTE: venous thromboembolism.

Question 6.

General:

Avoid the use of abbreviations without them being first elaborated in full form. E.g. COPD, BMI, PaO2, CT etc.

Response

Thank you for the appropriate remark. We modified according to your suggestions the manuscript.

Reviewer 3 Report

The manuscript is very well written. I have the following minor recommendations for helping improve this manuscript:

  1. In Conclusion, please modify the following sentences:
  2. Original sentence: In the lack of validated diagnostic algorithm or guidelines for PTE in TB patients, all TB patients with extensive disease should undergo D-dimer testing, and PTE should be suspected and investigated with CTPA whenever serum D-dimer levels are elevated.

Modification recommended: Given the lack of validated diagnostic algorithms or guidelines for PTE in TB patients, D-dimer should be tested for all TB patients with extensive disease, and PTE should be suspected and investigated with CTPA whenever serum D-dimer levels are elevated.

  1. Original sentence: Further researches are needed to evaluate all risk factors that may increase the risk of PTE in TB patients and to develop validated diagnostic algorithms to arrive at a timely diagnosis and treatment of PTE in these patients.

Modification recommended: Further research is needed to evaluate all risk factors that may increase the risk of PTE in TB patients and to develop validated diagnostic algorithms to arrive at a timely diagnosis and treatment of PTE in these patients.

  1. In Discussion, please modify the following sentence:
  2. Original sentence: We found a raising incidence of PTE from 0.6% in 2016-2019 period to 4.6% in 2020-2021 period, and we speculate that the increasing in PTE prevalence in the latest two years period could be associated with the increased severity of TB cases.

Modification recommended: We found a rise in incidence of PTE from 0.6% in 2016-2019 period to 4.6% in 2020-2021 period, and we speculate that the increase in PTE prevalence in the past two years period could be associated with the increased severity of TB cases.

Author Response

REVIEWER 3

Comments and Suggestions for Authors

The manuscript is very well written. I have the following minor recommendations for helping improve this manuscript:

Response

We thank you very much for the encouraging feedback on our manuscript. Below the answers for your interesting and appropriate questions.

Question 1

In Conclusion, please modify the following sentences:

2.Original sentence: In the lack of validated diagnostic algorithm or guidelines for PTE in TB patients, all TB patients with extensive disease should undergo D-dimer testing, and PTE should be suspected and investigated with CTPA whenever serum D-dimer levels are elevated.

Modification recommended: Given the lack of validated diagnostic algorithms or guidelines for PTE in TB patients, D-dimer should be tested for all TB patients with extensive disease, and PTE should be suspected and investigated with CTPA whenever serum D-dimer levels are elevated.

Response 1

Thank you for the suggestion.

We modified manuscript at page 7 lines 291-294 as follow:

Given the lack of validated diagnostic algorithms or guidelines for PTE in TB patients, we suggest to perform systematic measurements of d-dimers in all TB patients with extensive disease. PTE should be suspected and investigated with imaging tests whenever serum D-dimer levels are elevated

Question 2

1.Original sentence: Further researches are needed to evaluate all risk factors that may increase the risk of PTE in TB patients and to develop validated diagnostic algorithms to arrive at a timely diagnosis and treatment of PTE in these patients.

Modification recommended: Further research is needed to evaluate all risk factors that may increase the risk of PTE in TB patients and to develop validated diagnostic algorithms to arrive at a timely diagnosis and treatment of PTE in these patients.

Response 2

Thank you for the suggestion.

We accordingly modified manuscript at page 7 lines 295-297.

Question 3

In Discussion, please modify the following sentence:

Original sentence: We found a raising incidence of PTE from 0.6% in 2016-2019 period to 4.6% in 2020-2021 period, and we speculate that the increasing in PTE prevalence in the latest two years period could be associated with the increased severity of TB cases.

Modification recommended: We found a rise in incidence of PTE from 0.6% in 2016-2019 period to 4.6% in 2020-2021 period, and we speculate that the increase in PTE prevalence in the past two years period could be associated with the increased severity of TB cases.

Response 3

Thank you for the suggestion.

We accordingly modified manuscript at page 6 lines 217 - 219.

Round 2

Reviewer 1 Report

The suggested improvements have been carried out.

In the newly written parts there are some small errors (high... high, d-dimer....) to be corrected. Otherwise acceptable.

Reviewer 2 Report

I agree with the corrections made by the authors.